# Development and Investigation of High-Temperature Ultrasonic Measurement Transducers Resistant to Multiple Heating–Cooling Cycles

**DOI:** 10.3390/s23041866

**Published:** 2023-02-07

**Authors:** Vaida Vaskeliene, Reimondas Sliteris, Rymantas Jonas Kazys, Egidijus Zukauskas, Liudas Mazeika

**Affiliations:** Ultrasound Research Institute, Kaunas University of Technology, Barsausko St. 59, LT-51368 Kaunas, Lithuania

**Keywords:** high-temperature ultrasonic transducer, PZT, silver electrode, copper, modelling, heating–cooling cycles

## Abstract

Usually for non-destructive testing at high temperatures, ultrasonic transducers made of PZT and silver electrodes are used, but this could lead to damage to or malfunction of the ultrasonic transducer due to poor adhesion between PZT and silver. Soldering is one of the most common types of bonding used for individual parts of ultrasonic transducers (protector, backing, matching layer, etc.), but silver should be protected using additional metal layers (copper) due to its solubility in solder. A mathematical modelling could help to predict if an ultrasonic transducer was manufactured well and if it could operate up to 225 °C. The observed von Mises stresses were very high and concentrated in metal layers (silver and copper), which could lead to disbonding under long-term cyclic temperature loads. This paper presents a multilayer ultrasonic transducer (PZT, silver electrodes, copper layers, backing), which was heated evenly from room temperature to 225 °C and then cooled down. In the B-scan, it was observed that the amplitude of the reflected signal from the bottom of the sample decreased with an increase in temperature. However, after six heating–cooling cycles, the results repeated themselves and no signs of fatigue were noticed. This ultrasonic transducer was well manufactured and could be used for non-destructive testing when the environment temperature changes in cycles up to 225 °C.

## 1. Introduction

Ultrasonic transducers used for measurement or non-destructive testing purposes in general are like sandwiches of different components, namely a piezo element, electrodes, matching layer, and backing. The adhesion between those layers is the most important parameter defining the survivability of a transducer during exploitation in high-temperature conditions. Usually, commercially available piezoelectric ceramic elements are coated with silver or gold electrodes [1,2,3]. Although silver is a good choice for high-temperature applications due to its high electrical and thermal conductivity, this metal is expensive, so it could be replaced with copper for lower temperature applications [4,5,6,7]. On the other hand, silver is a soft metal, which causes problems during soldering due to its reactivity with sulfuric molecules in the air and solubility in the solder. Additionally, our experience showed that adhesion between commercially available lead zirconate titanate ceramic (PZT) elements and applied silver electrodes is poor. The reason that the electro-mechanical coupling of the whole ultrasonic transducer is poor is due to the porosity of the silver layer and/or the PZT ceramic element, which leads to variations in the thickness of the ceramic disk–electrode joint. Therefore, when operating in a wide range of temperatures, this bond can be easily destroyed due to different deformations in individual parts of ultrasonic transducers caused by thermal expansion [4,8,9,10]. When comparing different types of piezoelectric materials (lead zirconate titanate, lithium niobate, bismuth titanate, gallium orthophosphate, aluminum nitride, etc.) that could be used for non-destructive testing at high temperatures, PZT has the lowest Curie point (160–365 °C); however, in our case, it is sufficient. Other piezoelectric materials could work in temperatures up to 1100 °C, but they possess lower electromechanical coupling factors and, due to different compositions of the material, the joining method should be essentially different. Pz27 is a soft PZT material, whose main characteristics are as follows: Curie point (350 °C); recommended working temperature range (up to 250 °C); high electromechanical coupling factors (*K_p_* = 0.59, *K_t_* = 0.47); good piezoelectric properties (*d_33_* = 440 pC/N); stable performance; and low aging rate, which is important for our experiments. The manufacturer states that a piezoelectric charge coefficient (*-d_31_*) of Pz27 slightly increases when the temperature rises (25–225 °C). Therefore, it also demonstrates good stability of performance in temperature ranges corresponding to our tasks. Based on its known properties and our experience with it, Pz27 was chosen as the piezoelectric material for our developed ultrasonic transducer [2,11,12].

Another important question is how to join reliably individual parts of an ultrasonic transducer (protector, backing, matching layer, etc.). For this purpose, different techniques are used, such as soldering, welding, gluing, etc. [11]. To protect the silver electrodes from temperature effects, such as solubility in solder, an additional metal layer can be introduced to the protector by soldering before bonding. In a wide temperature range, any part of the ultrasonic transducer can be damaged by bad adhesion between layers that are not strong enough to obtain stable ultrasonic signals. All parts and metal coatings of the ultrasonic transducer must be acoustically compatible and must preferably possess similar thermal expansion coefficients. The selection of such elements should be considered wisely, and all parts should be properly connected in order to not change the characteristics of the ultrasonic measurement transducer and to avoid deformation or even cracking at high temperatures (>200 °C) [4,11,13]. Alternatives to soldering or welding silver include dry contact bonding, diffusion bonding, or, for a more advanced technique, transient liquid phase diffusion bonding [14,15] could be used to bond metal layers and ceramics. However, these techniques require advanced technical skills and sophisticated equipment to work at high pressures and or high temperatures [11,13,14,15,16,17].

To predict the performance of an ultrasonic transducer at elevated temperatures, and to predict possible failures, mathematical modelling can be applied. For this purpose, a finite element method could be used to calculate and predict possible stresses and deformations in a structure of the transducer sandwich at different temperatures [18,19]. Mathematical modelling software COMSOL could be used for this purpose. This software yields predictions of materials’ behavior in a wide temperature range based on their coefficients of thermal expansion. Additionally, it yields the determination of the possible occurrence of different types of stresses and deformations, which could lead to the malfunction of the ultrasonic transducer [20]. However, we could not find any publications related to the investigation of deformations of ultrasonic transducers at different temperatures.

The objective of this paper was the development and investigation of the ultrasonic transducer performance in a temperature range of up to 225 °C during multiple heating and cooling cycles, as well as the identification of key issues that determine the survivability of the probe.

In this paper, the structure of the high-temperature transducer is presented first, paying primary attention to the improvement of the adhesion of the electrodes to the piezoceramic element. For that purpose, the silver electrodes are coated by copper and the techniques used to do so are demonstrated. In the following section, the expected stresses in the structure of the ultrasonic transducer are assessed using numerical modelling. The final section is devoted to the experimental investigation of the performance of the developed transducer during temperature increase/decrease cycles.

## 2. Materials and Methods

### 2.1. Design of an Ultrasonic Transducer and Improvement of Adhesion between Electrodes and Piezoceramic

For this developed ultrasonic transducer, a commercially available Soft PZT (Navy II) Type Pz27 piezo element (diameter, 13 mm; thickness, 1 mm; frequency, 2 MHz), with two already applied silver electrodes (~10 µm thickness), was used [12]. On both silver electrodes, a copper layer (5 µm thickness) was applied by electroplating. As an additional layer, a thin copper foil (copper front layer) was soldered to the silver electrode of the front transducer covered with a thin layer of copper (applied by electroplating), using tin (due to its thermal and ultrasonic properties at high temperatures) as the solder. The thickness of the tin layer should be minimal (thickness of less than 0.1 λ) so as to influence the change in frequency and other characteristics of this developed transducer as little as possible. The copper foil (thickness, 0.2 mm) was used for mechanical strength and to connect the transducer electrodes. The tin layer on the back copper-clad electrode had a roughened surface to increase the surface area and improve the adhesion of the backing (Careco epoxy stick) and, at the same time, the acoustic contact over a wide temperature range. The schematic image of this ultrasonic transducer is shown in Figure 1.

Additional metal layers on the silver electrodes were placed to prevent silver electrodes from possible damage during silver soldering because of the chemical and physical properties of silver [4]. Copper was chosen for electroplating on silver because of their similar properties (melting point, coefficient of thermal expansion, Poisson’s ratio) and because of the possibility to use tin for solder, which has similar properties to both mentioned metals. It is also easier to solder and to bond to additional parts of an ultrasonic transducer. The most important properties of silver, copper, and tin for upcoming processes are presented in Table 1.

**Table 1 sensors-23-01866-t001:** Properties of silver, copper, and tin.

Properties	Silver [6]	Copper [21]	Tin [22]
Melting point, °C	961.93	1083.2–1083.6	231.968
Coefficient of thermal expansion, µm/m-°C	19.9 (at 250 °C)	18.5 (at 250 °C)	30.0–40.4 (at 200 °C)
Specific heat capacity, J/g-°C	0.234	0.385	0.213
Thermal conductivity, W/m-K	419	385	63.2
Modulus of elasticity, GPa	76	110	41.6–44.3
Poisson’s ratio	0.370	0.343	0.330

To properly conduct copper electroplating on silver, the surface of the silver should be prepared and perfectly cleaned. First, acetone was used to degrease the surface. Then, for cleaning, a 10 g/L sodium carbonate (Na_2_CO_3_) solution and ultrasonic bath were used. An electroplating solution was made by mixing copper sulfate (150–250 g/L) and sulfuric acid (15–60 g/L). A 1–10 A/dm^2^ current (Figure 2) was applied for several minutes depending on the desired thickness [23,24,25]. All solutions were made of chemically pure reagents and bi-distilled water.

In Figure 2, a scheme of copper electroplating is shown. A piezoceramic element with silver electrodes was wrapped with copper wire, which was connected to the negative electrode by a clip. A copper plate was put inside the beaker to cover the walls of the beaker from the inside and to create a uniform electric field. The copper plate was connected to the positive electrode by another clip. Then, the beaker was filled with a mix of copper sulfate and sulfuric acid, and the piezoceramic element was immersed in the solution at the center of the copper plate. Finally, the current was applied to the solution.

### 2.2. Numerical Modelling of Thermal Stress in Coating of Piezoelectric Ceramic

The proposed structure consists of different layers whose materials have a wide range of thermal expansion coefficients. Under the influence of high temperature, the materials of the structure’s layers expand differently, and this causes thermal stress in the structure. The easiest way to evaluate stresses in such a multilayer structure is through numerical modelling using the finite element technique. Numerical modelling of thermal stress in the coating of the piezoelectric ceramic disc was performed using COMSOL finite elements software. Due to cylindrical structure of the object under investigation, the model was simplified to a 2D axisymmetric model. A graphical representation of the finite element model and its dimensions is presented in Figure 3. In this numerical investigation, the piezoelectric effect of the piezoelectric ceramic was not considered. To simplify modelling, the assumption that the piezoelectric ceramic was isotropic was made. The assumption that the adhesion between layers was perfect was also made. The mechanical and thermal properties of the materials used during modelling are presented in Table 2. Convective heat-transfer boundary conditions with a heat-transfer coefficient of 10 W/(m^2^·°C) were applied to the external edges of the object under investigation. Stationary structural analyses were performed for different ambient temperatures from 40 °C to 200 °C in increments of 40 °C.

**Table 2 sensors-23-01866-t002:** Material properties used in finite element modelling.

Property	Pz27 [26,27]	Silver [6]	Copper [20]
Density, kg/m^3^	7700	10,491	8960
Young’s modulus, GPa	60	76	110
Poisson’s ratio	0.31	0.37	0.35
Thermal conductivity, W/m-°C	1.3	419	400
Specific heat, J/(kg·°C)	440	234	385
Thermal expansion coefficient 1/°C	2·10^−6^	19.9·10^−6^	17·10^−6^

**Figure 3 sensors-23-01866-f003:**
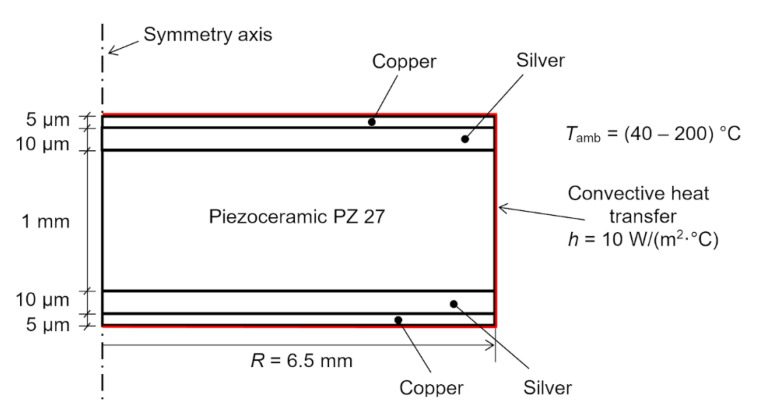
Graphical representation of the finite element model. Dimensions are not to scale.

Thermal stresses in the investigated structure were evaluated in a term of von Mises stress. The von Mises stress is scalar quantity and criterion of yield very often used for metals. If the von Mises stress is greater than the yield strength of the material, then the material will yield. The distribution of von Mises stress in the piezoelectric ceramic disc coated with silver and a copper layer at 200 °C is presented in Figure 4.

Due to significant differences in the thermal expansion coefficients of the piezoelectric ceramic and coated layers, the obtained stresses are very high and concentrated in the coated layers. The resulting stresses greatly exceed the yield strength of silver and copper, which are 45 MPa for silver and 33 MPa for copper. Stress distributions in the silver layer along the radius of the piezoelectric ceramic at different temperatures are shown in Figure 5. It is seen that only at a temperature of 40 °C does the von Mises stress not exceed the yield strength of the material. At higher temperatures, damage to the coated layers may occur. On the other hand, the current model does not consider bonding between layers. Due to a big difference in the thermal expansion coefficients of the materials, the disbonding of the coated layers of the piezoelectric ceramic may occur. Both phenomena (exceeding yield strength and disbonding the layers) negatively affect the performance of the ultrasonic transducer. There is a high probability that under long-term cyclic temperature loads, the coated layers of the piezoelectric ceramic will disbond and the ultrasonic transducer will stop working.

### 2.3. Set-Up of Experiment

In order to test the performance of the developed ultrasonic transducer during multiple heating–cooling cycles, the experimental set-up needs to be prepared. The signals were measured on a carbon steel block (height, 100 mm; width, 100 mm; length, 150 mm) to which the transducer was attached using a specially developed holder ensuring a constant pressure force throughout the duration of the experiments (Figure 6). The coupling liquid, the Sono 600 high-temperature coupling gel (Magnaflux, Glenview, IL, USA) was used between the developed ultrasonic transducer and the carbon steel block. The ultrasonic system ULTRALAB-2 (developed by the Ultrasound Institute of Kaunas University of Technology, Lithuania) was used to generate and record signals. The two-channel ultrasonic measurement system consists of a high-voltage generator, a low noise amplifier, and an analogue-to-digital converter. The maximal output voltage of the generator was 750 V. The amplifier gain can be changed from 10 dB to 80 dB. The low noise 20 dB preamplifier was connected directly to the transducer in order to improve the signal-to-noise ratio. The whole system was controlled by a personal computer. Experimental signals were collected and stored in the PC. Additionally, there was a possibility of the real-time presentation of B-scan and C-scan images. The transducer was excited by the rectangular impulse with an amplitude of 20 V and a duration of 250 ns, and the signals reflected by the backwall of the steel block were recorded with repeatability every 10 s throughout the experiments. To heat the test block with the attached transducer, the heat oven Binder ED-53 (Tuttlingen, Germany) was used. Two thermocouples were used to monitor the temperature of the oven and the test block. The first one (type K) was placed in the middle of the oven, and the second one was attached to the steel block. One of the thermocouples was connected to the second channel of the ULRALAB system and another (FLUKE 5608 type thermocouple, Everett, WA, USA) was connected to a separate temperature-measurement unit (FLUKE 1523 type reference thermometer, Everett, WA, USA). The schematic view of this experimental set-up is shown in Figure 6.

## 3. Results

The next stage of this research was to investigate the performance of the developed high-temperature ultrasonic transducer under six temperature variation cycles. Using the experimental set-up presented in Section 2.3 multiple heating–cooling cycles were performed. During the cycle, heating started at room temperature and continued to 225 °C. The heating stage took place for 8 h, after which heating was shut down and the process of cooling started. Cooling was also monitored for 8 h. The total duration of a single cycle was 16 h. The variations in air temperature inside the oven and test block are shown in Figure 7.

It was observed that in less than 1 h, the temperature inside the oven reached 200 °C, while the temperature of the sample was only ~80 °C. After 6 h of heating, the sample reached 200 °C and the temperature of the environment was ~225 °C. The temperature measurement of the sample and inside the oven stopped after 8 h of cooling. It can be noticed that, even after 8 h, the cooling temperatures of the test block and oven were still high at 55 °C and 40 °C, respectively. These results were crucial for this experiment because the carbon steel sample did not heat up as quickly as the environment and it was important to track the temperature changes inside the sample to be able to understand the changes in the signals.

The signals reflected from the bottom of the sample at different temperatures (25 °C, 130 °C, and 190 °C) and their spectra are presented in Figure 8a,b. As can be seen, the highest amplitude occurred at the lowest temperature (blue curve) and the lowest amplitude occurred at 190 °C (red curve). It can also be seen that the signals measured at high temperatures were delayed with respect to those measured at a cooler temperature. This is explained by the fact that the ultrasound propagation velocity in steel decreases with the increase in temperature. The total variation in the signal reflected by the bottom of the test block is presented in Figure 9 in B-scan-type images.

A frequency-based analysis of the received echo signals in the range of experimentally tested temperatures showed no influence of temperature on the shape of the frequency spectra of the received signals in general (Figure 8b). The temperature only affects the amplitude of the spectra components, which reduces with the increasing temperature. The negligible variations in the frequency spectra of signals are probably caused by changes in gap thickness between the transducer and the test sample.

A comparison of the transducer’s reverberation signals in the temperature range (Figure 9a) shows the reverberation “tail” growing just after excitation with the increase in temperature. The reflected signal decreased at the same time. This can be explained by the deterioration of the acoustical contact between the ultrasonic transducer and the test sample with the increase in temperature. It caused the reverberation “tail” to grow, as well as a weaker acoustic contact and, as a result, a weaker reflected signal. The decrease in acoustical contact could be explained by bigger displacements of the transducer edge caused by higher stresses at higher temperatures as it follows the presented finite element modelling. As a result of such deformation of the front surface of the transducer, the thickness of the liquid contact layer increased, leading to a reduction in the amplitude of the received ultrasonic signal reflected from the bottom of the test sample.

In the B-scan, a dependence between the amplitude of the signal and the temperature is presented, where *T_e_* is the time at which the measurements were performed, and *t* is the propagation time of the reflected ultrasonic signal. Before heating the sample, the amplitude of the reflected signal from the bottom of the sample was at its highest (7 × 10^−3^ V). The more the temperature increased, the more the amplitude of the reflected signal decreased. The amplitude of the reflection was at its lowest (2 × 10^−3^ V) after 6–8 h of heating when the temperature of the sample was 200 °C. When the carbon steel cooled to 160 °C, the amplitude of the reflected signal began to increase. At the end of this experiment (after 8 h of cooling down), it was almost the same as it was at the beginning because the temperature of the sample was higher than it was before heating (55 °C and 25 °C, respectively).

According to the temperature curve of the sample (Figure 7) after 6–8 h of heating, the temperature of the carbon steel was stable (200 °C) but in Figure 10, it can be seen that the normalized amplitude of the reflected signal decreased from −11 dB to −13 dB. This could be explained by the measurement of the temperature of the sample because it was measured at the surface, and the sample heated up from the surface to its center. The ultrasonic waves propagated throughout the whole volume of the sample, so despite the constant surface temperature it still increased inside the specimen, and then a decrease in the amplitude of the reflected signal was observed. As the sample cooled down, the amplitude of the signal increased again and almost returned to the initial value.

The observed slight hysteresis in Figure 10b can be explained by the fact that during a heating cycle, the temperature of the ultrasonic transducer rose faster than the temperature of the steel test sample. However, during the cooling cycle, its temperature was similar to the temperature of the test sample.

After six repeated cycles of heating and cooling the sample, the results repeated themselves, and no signs of fatigue were noticed, so it could be concluded that there were no residual deformations. This developed ultrasonic transducer was well manufactured, and could be used for non-destructive testing when the environmental temperature changes in cycles up to 225 °C.

## 4. Discussion

The developed high-temperature ultrasonic transducer may operate up to 225 °C and is resistant to multiple heating–cooling cycles, which is characteristic for many non-destructive testing applications. Such a result was achieved by improving the adhesion quality between internal components of the transducer by means of additional copper layers. The prediction modelling of the thermal expansion of the developed ultrasonic transducer demonstrated that very strong deformations and stresses could be expected at the edges of the sandwich structure, which can lead to the delamination of the electrodes during heating–cooling cycles. However, the developed technology of electroplating electrodes yielded the softening of stresses arising at the edges of the piezo element and electrodes. In such a case, the survivability of the ultrasonic transducer during multiple heating–cooling cycles increased.

On the other hand, the experiment revealed that the amplitude of the signals propagated through the steel block possessed significant amplitude variations. During heating up to 225 °C, it dropped by 12 dB, but it completely recovered when the transducer cooled down to the room temperature. There is no clear explanation for this, but possible reasons include a variation in the properties of the coupling liquid, different diffraction conditions of the waves propagating in the steel block with temperature gradients, and variations in the electromechanical coefficients of the piezoceramic. However, we think that variations in the electromechanical coefficient cannot significantly affect the amplitude of the reflected signal because our measurements of the electromechanical coefficient in a temperature range of up to 225 °C showed that they were insignificant. Nevertheless, the experiments demonstrated relatively good repeatability of the amplitude variations during different heating cycles. As such, it can be stated that the developed ultrasonic transducer demonstrated good survivability and performance at elevated temperatures.

## Figures and Tables

**Figure 1 sensors-23-01866-f001:**
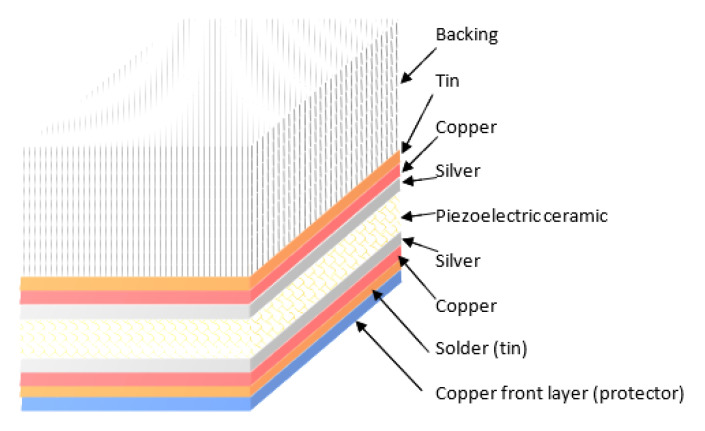
The developed ultrasonic transducer.

**Figure 2 sensors-23-01866-f002:**
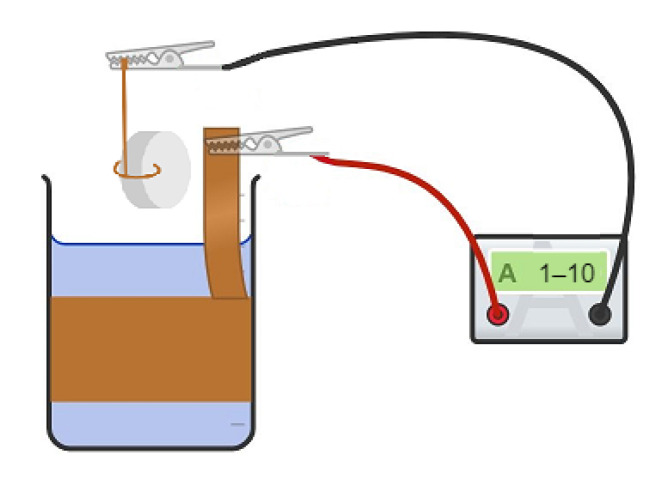
Scheme of copper electroplating.

**Figure 4 sensors-23-01866-f004:**
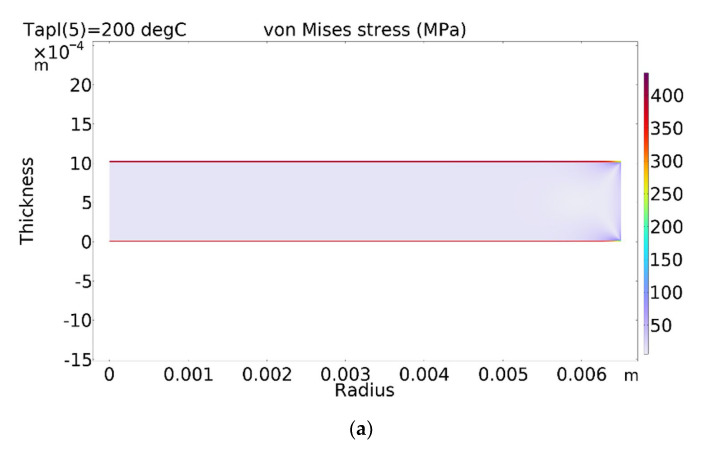
Distribution of von Mises stress in piezoelectric disc at 200 °C temperature: (**a**) full view; (**b**) zoomed edge of the top layer.

**Figure 5 sensors-23-01866-f005:**
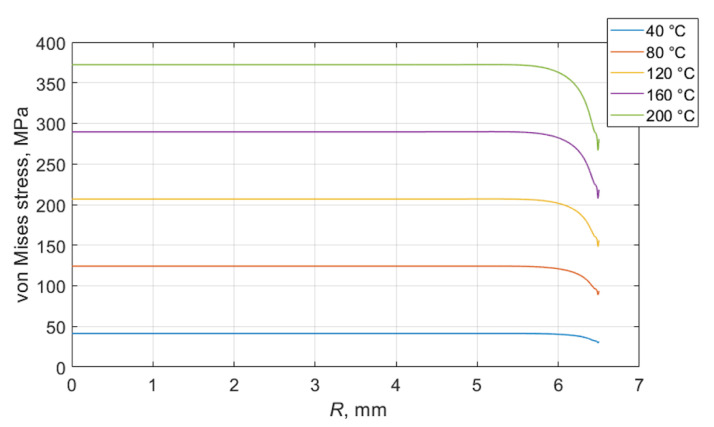
Distribution of von Mises stress in silver coating along the radius of piezoelectric disc.

**Figure 6 sensors-23-01866-f006:**
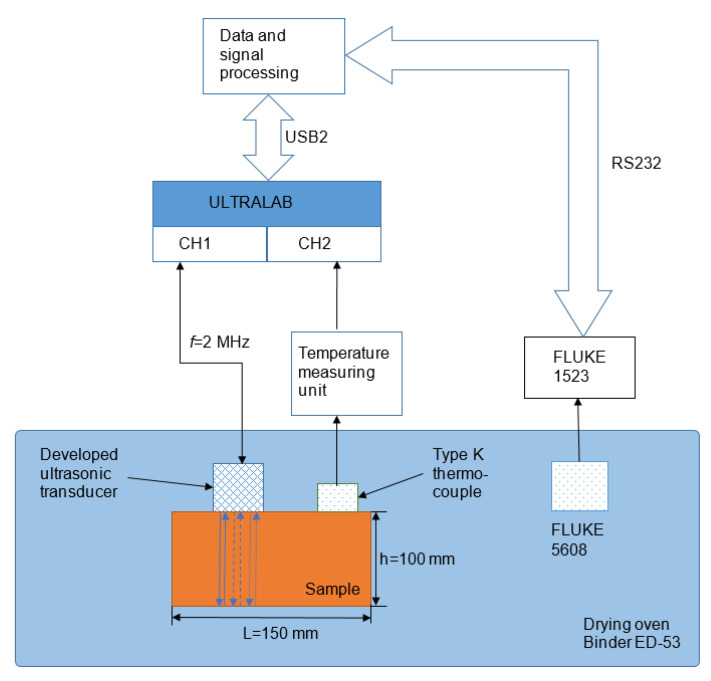
Set-up of experiment.

**Figure 7 sensors-23-01866-f007:**
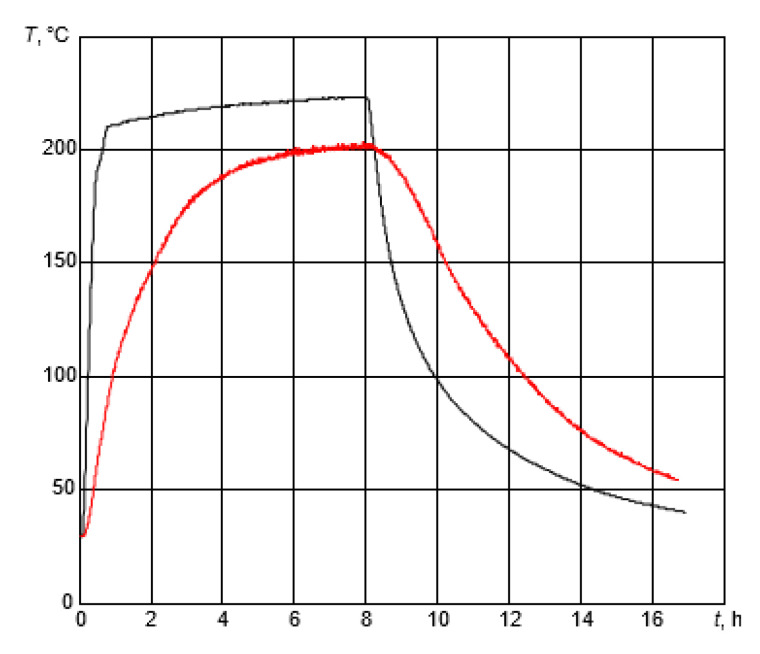
A thermal test of the developed ultrasonic transducer at 25–225 °C. Black: the temperature inside the drying oven; red: the temperature of the sample.

**Figure 8 sensors-23-01866-f008:**
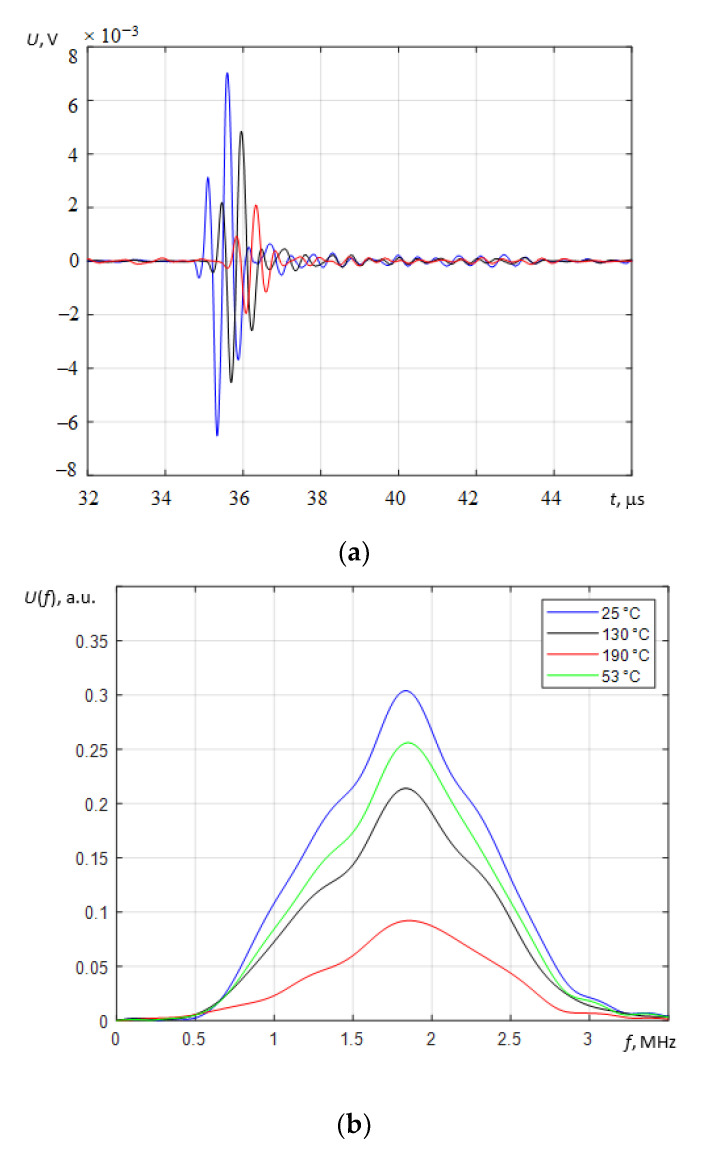
The reflected signals from the bottom of the test sample (**a**) and their spectra (**b**) at different temperatures of the sample. Blue: 25 °C; black: 130 °C; red: 190 °C; green: 53 °C (during cooling).

**Figure 9 sensors-23-01866-f009:**
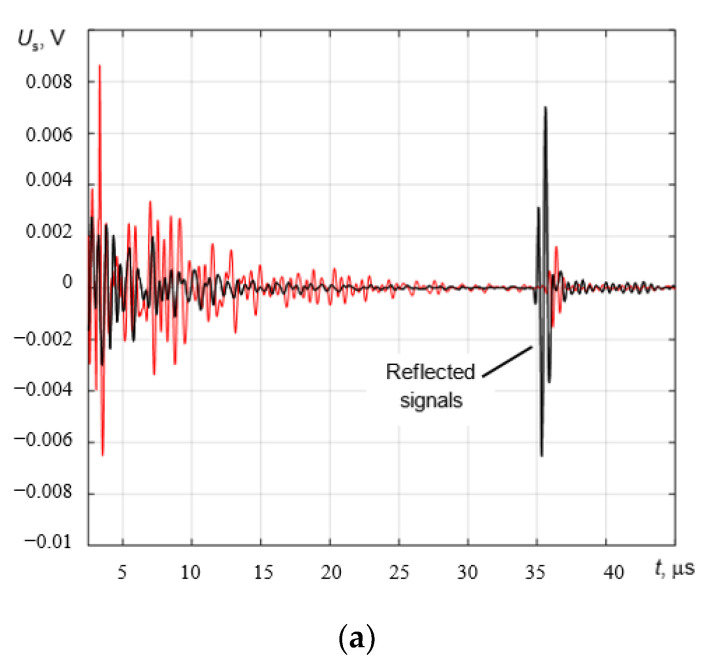
The recorded ultrasonic signals at different temperatures: (**a**) reverberations after excitation of the transducer and reflected signals; (**b**) zoomed-in view of the B-scan.

**Figure 10 sensors-23-01866-f010:**
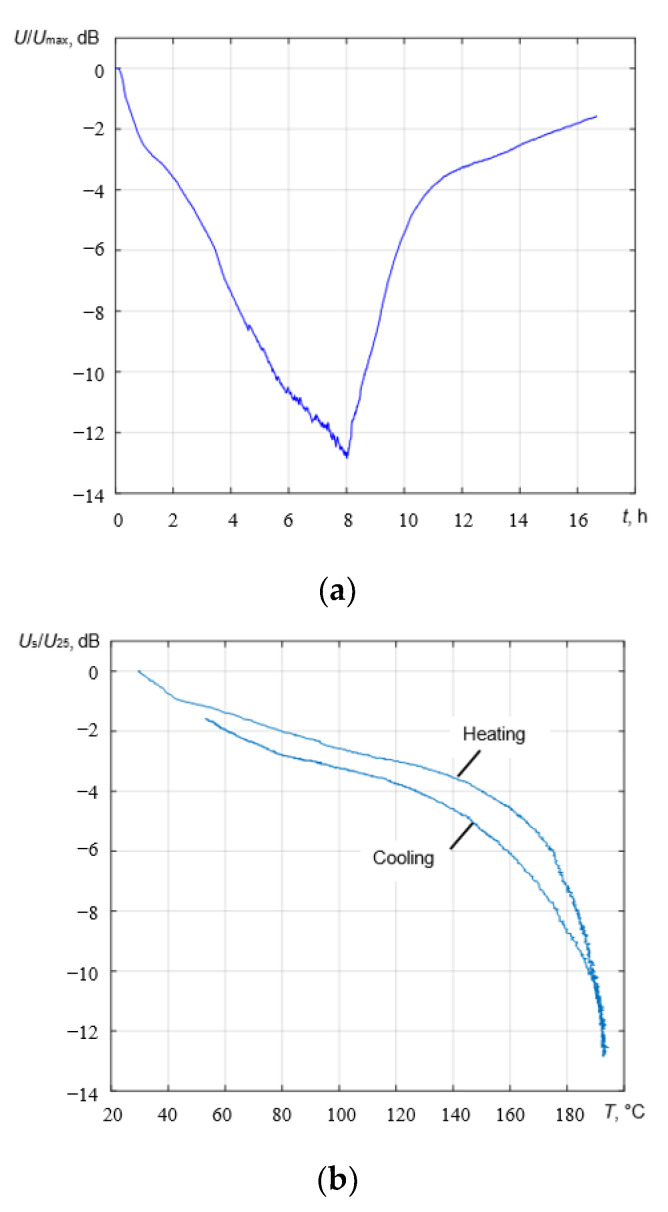
Signal amplitude. (**a**) Normalized amplitude of the reflected signal; (**b**) signal amplitude versus test block temperature.

## Data Availability

The data presented in this study are available on request from the corresponding author.

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
