# Peer review of "Development and Investigation of High-Temperature Ultrasonic Measurement Transducers Resistant to Multiple Heating–Cooling Cycles"

_sensors, 2023, doi:10.3390/s23041866_

Round 1

Reviewer 1 Report

This is a good piece of work on the design of an ultrasonic sensor that can withstand repeated thermal cycling without apparent degradation of quality. The combination of experimental results, intelligent design, and numerical modelling will make this a useful paper for other researchers.

I note that the English does need some polishing to remove awkward phrasing and the occasional ambiguous/unclear statement, but this is not a major concern. Specific comments follow:

1.     Lines 86-89. This paragraph was confusing. First you state that a copper layer was applied to both electrodes by electroplating. However, you then state that a copper foil was applied by soldering. So I am confused: Are you applying a thin foil, or electroplating? Or are you talking about two different copper layers?

2.     Figure 1: There are a few layers of the transducer that have not been adequately explained. What is the “buffer” layer? You also state that an outer copper layer acts as a “damper” – is that correct? Copper is not known for its damping properties.

3.     Line 102: I do not understand your reference to tin “plates”. I understand that you are using tin for solder, but the word “plate” implies a relatively thick manufactured structure at least a few mm thick.

4.     Table 1. Something is not quite right with Table 1. For an isotropic material,

G = E/2/(1+nu), where “nu” is Poisson’s ratio. Your data are consistent with that equation for the case of silver, but are seriously off for the copper. Please check your value of G, E, and nu for copper.

5.     Lines 116-123: Won’t your electroplating scheme lead to a short-circuit between the two poles of your piezoelectric disc?

6.     Lines 169-175: We need more information on the amount of applied stress between transducer and steel block. Also, what sort of surface preparation was applied to the two contacting surfaces?

7.     Line 239: It would be helpful to see a graph that shows signal amplitude versus block temperature, both as the block temperature is increasing and again while it is decreasing. In particular, I would be interested in seeing if there is any hysteresis between the graphs of increasing and decreasing temperature – this might give extra insight as to what causes the temperature dependence of signal amplitude.

8.     Line 262: It is helpful that you are able to eliminate variable electromechanical efficiency as the cause of the temperature dependence of your results. However, that still leaves me uncertain as to whether the temperature dependence is cause by the couplant (variable viscosity with temperature), or something to do with your transducer design. Have you considered trying an alternative coupling technique on a single test transducer (e.g., firmly bonding the transducer to the steel block) and then repeating the experiment?

Reviewer 2 Report

This manuscript discussed the temperature cycling reliability of high temperature ultrasonic measurement transducer manufactured by soldering the layers. There are significant concerns for this manuscript. So the reviewer suggests rejection. 

1. Introduction: Any proof of "the porosity of the silver layer and/or PZT ceramic elements"?

2. Session 2.1: "The thickness of the tin layer should be minimal". So how much is the thickness exactly? The thickness of the solder layer can significantly influence the thermal cycling reliability of the bonding. 

3. In fact Ag can also react with Sn, the most common intermetallic compound (IMC) is Ag3Sn. So, why the Cu layer is necessary?

4. The melting point of Sn is 231.9 C. 225 C is very close to it. Sn can react very quickly with Cu or Ag. There will be very fast IMC generation at the interface. Eventually, there will be very thick IMC generated. The COMSOL simulation will mean less. The property of the device will also be influenced. The authors cannot claim "After several repeated cycles of heating and cooling the sample, the results repeated themselves, and no signs of fatigue were noticed, so it could be concluded that there were no permanent deformations". 

Reviewer 3 Report

Authors showed new high tempeature multi-layer structures of the ultrasonic transducers. However, there are some necessary references and additional analysis for the experiments missing. There are no big English grammar issues. Therefore, authors need to revise the manuscript according to the comments as below.

1. Labels of Figures 4 and 5 are small so authors had better increase the font sizes of the x- and y-axes.

2. In Figure 9, what are the labels of x- and y-axes of t ? Authors used the same time (t) but they have different definitions.

3. Data availability section is missing.

4. Authors should provid ref. (Ultrasonic transducers used for measurement~) with ref.(https://www.sciencedirect.com/science/article/abs/pii/S0263224116306157)

5. Authors showed some time-based echo signal. How about the frequency-based echo signal ?

6. Please correct font of the 7x10-3 V.

7. Authors had better compare your measured results with previous studies. If possible, please provide the Table with some references (which mentioned in the Introduction section).

8. In Line 258, please correct 12dB to 12 dB. Please check others.

9. Please describe the system ULTRALAB-2 in detail.

10. In Figure 10, is there any saturation point after 16 hour ?
